# Simple and Accurate HPTLC-Densitometric Method for Assay of Nandrolone Decanoate in Pharmaceutical Formulation

**DOI:** 10.3390/molecules24030435

**Published:** 2019-01-25

**Authors:** Małgorzata Dołowy, Alina Pyka-Pająk, Josef Jampílek

**Affiliations:** 1Department of Analytical Chemistry, School of Pharmacy with the Division of Laboratory Medicine in Sosnowiec, Medical University of Silesia in Katowice, Jagiellońska 4, 41-200 Sosnowiec, Poland; apyka@sum.edu.pl; 2Department of Pharmaceutical Chemistry, Faculty of Pharmacy, Comenius University in Bratislava, Odbojárov 10, 832 32 Bratislava, Slovak Republic; 3Regional Centre of Advanced Technologies and Materials, Palacky University, Slechtitelu 27, 783 71 Olomouc, Czech Republic

**Keywords:** steroids, nandrolone decanoate, HPTLC, densitometry

## Abstract

This study reports the development and validation of a new, simple, and accurate high-performance thin-layer chromatography (HPTLC)-densitomeric method for the determination of nandrolone decanoate in a commercially available injection formulation. Chromatographic analysis was performed on glass CN modified silica gel 60F_254_ plates developed using n-hexane-ethyl acetate in volume ratio 42.5:7.5 as the mobile phase. Densitometric scanning was carried out at the wavelength of 245 nm. This chromatographic system gave compact spot and a symmetrical peak of nandrolone decanoate with retardation factor (R_F_) value at 0.57 (±0.02). The linearity of this method with the high correlation coefficient of calibration plot ranges from 0.780 to 12.500 μg/spot. The developed method is characterized by good precision (coefficient of variation CV < 2%) and high accuracy close to 100.3% (R = 99.0%). Values of limits of detection and quantification equal to 0.231 and 0.700 μg/spot, respectively, confirm the sensitivity of the developed method. The analysis of the pharmaceutical formulation of nandrolone decanoate indicates drug content of 50.5 mg/mL and 101.0% in relation to the label claim. This is in good agreement with the recommendation of the International Council for Harmonisation (ICH) guidelines as well as the pharmacopoeial requirements. The low CV value (<1%) of nandrolone decanoate content in the tested injection formulation confirms the suitability of the proposed HPTLC-densitometric method for routine control of this compound in examined pharmaceuticals.

## 1. Introduction

Nandrolone (19-nortestosterone) represents an important class of anabolic-androgenic steroids (AAS) [1,2]. It is widely used as a therapeutic agent in the form of esters such as nandrolone decanoate (ND) and nandrolone phenylpropionate (NPP) in the treatment of anemias and osteoporosis in postmenopausal women [2]. In addition to clinical uses, nandrolone and its derivatives have been applied as doping agents in sport by various athletes in order to improve the performance of their muscles, and it can be successfully used to increase feeding efficiency in animals. For this reason, drugs containing nandrolone are controlled substances. Similarly, as in the case of other AAS, the adverse effect of nandrolone and its metabolite residue (e.g., accumulated in meat and other foods) may include liver diseases, cardiovascular damage, and other organism dysfunctions. Because nandrolone is administered by injection into muscle to achieve a suitable effect, it is commercially available in the form of a solution for injection.

A survey of the literature reveals that various analytical procedures have been described for the determination of nandrolone residue and its metabolites in biological samples, for example, gas chromatography coupled to mass spectrometry (GC-MS) and high-performance liquid chromatography (HPLC) combined with UV-Vis spectrometry [3,4,5,6]. Gas chromatographic analysis of nandrolone in urine samples is sensitive and highly efficient but needs previous conversion into trimethylsilyl derivatives of nandrolone (TMS derivatives) [4]. In another work, Jiang and coauthors described the GC-MS analysis of nandrolone residue in animal tissues [5]. The sample extracts were derivatized previously with heptafluorobutyric anhydride (HFBA) and then were subjected to GC-MS study [5]. This method can be useful in case of nandrolone doping control in husbandry animals.

The next author, Mukherjee et al., has reported a reversed phase high-performance liquid chromatography (RP-HPLC) method for the determination of nandrolone phenylpropionate in bulk drugs and pharmaceutical formulations [6]. The chromatographic system consisted of a CN column and the mobile phase of phosphate buffer-acetonitrile (50:50, *v/v*). UV-Vis detection was performed at λ = 240 nm [6]. Our previous studies and the papers prepared by other researchers confirm also the utility of thin-layer chromatography (TLC) for the purpose of separation and quantitative analysis of various steroids, including selected anabolic-androgenic agents [7,8,9,10,11,12,13,14,15,16,17], because of its simplicity, low cost, and relatively low detection limit compared to other more modern chromatographic methods, such HPLC with different detection systems. Importantly, it does not need any tedious cleanup of the sample. The comparison of different chromatographic techniques such as TLC, as well as OPLC (over-pressured layer chromatography), methods in the purity control of nandrolone was reported by Bagocsi et al. [16]. It was stated that thin-layer chromatography was not a selective method in this study, and the applied OPLC analysis required multiple development of chromatographic plates and the use of visualizing reagent in the form of sulfuric acid. The limit of detection was ≤0.01 µg [16]. To the best of our knowledge, no work has been published concerning the TLC/HPTLC-densitometric analysis of nandrolone decanoate in injection formulations.

In continuation of our previous studies on the development of a HPTLC-densitometric method for analysis of steroids in pharmaceutical formulations [7,8,9,10,11], this work describes a rapid and accurate procedure suitable for effective quantitative assay based on the conventional high performance thin-layer chromatographic method in combination with densitometry (HPTLC-densitometry) for quantitative determination of nandrolone decanoate in bulk drugs, as well as the commercially available injection formulation. The proposed method was validated according to the guidelines of the International Council for Harmonisation of Technical Requirements for Pharmaceuticals for Human Use (ICH) [18].

## 2. Results and Discussion

### 2.1. Method Development

Different mixtures of organic solvents, e.g., acetone-n-hexane and n-hexane-ethyl acetate, in various volume compositions and variety of chromatographic plates precoated with non-modified silica gel 60F_254_, as well as CN modified silica gel 60F_254_, were used in this study.

Finally, the mobile phase consisted of n-hexane-ethyl acetate in volume ratio 42.5:7.5 and glass plates precoated with the modified CN silica gel gave compact spot as well as a symmetrical peak with acceptable retardation factor (R_F_) value at 0.57 (±0.02), see Figure 1. The bands of the sample (5 μL) and the standard solution at concentration of 0.50 μg/μL each were applied 10 mm from the lower edge of the plate as 6-mm wide bands. The chromatographic plates were developed in the chromatographic chamber, which was previously saturated with the mobile phase (50 mL) for 20 min, at the distance of 7.0 cm at room temperature. The developed plates were left to dry. Densitometric analysis of chromatograms was performed at 245 nm (±1 nm) using a TLC 3 Scanner. The wavelength was selected based on the maximum absorbance of UV-Vis spectra registered directly from the chromatogram in the range of 200–800 nm, see Figure 2.

### 2.2. Validation Results

The proposed HPTLC-densitometric method was validated in accordance with the ICH guidelines for specificity, linearity, limit of detection and quantification, precision, accuracy, and robustness in order to confirm the reliability of the obtained results [18].

#### 2.2.1. Specificity

The application of n-hexane-ethyl acetate in volume composition 42.5:7.5 and glass plates precoated with modified CN silica gel provided well-developed spots of nandrolone decanoate suitable for densitometric analysis. The specificity of the method was confirmed by comparing the R_F_ value and the spectra of the sample and the standard spot. As shown in Figure 3, a good agreement of R_F_ value of the standard and the pharmaceutical sample can be observed. On the obtained densitogram (from the pharmaceutical sample), no other peaks were found in addition to the peak coming from the active compound-nandrolone decanoate. The densitogram shows that there is no interference of excipients with studied nandrolone decanoate present in the examined pharmaceutical formulation. Hence, the developed HPTLC-method is specific and suitable for the determination of nandrolone decanoate in the tested pharmaceutical formulation.

#### 2.2.2. Linearity

The linearity of the developed method was evaluated by determining six working standard solutions in the wide range of nandrolone decanoate amount from 0.780 μg/spot to 12.500 μg/spot. The peak area (*y*-axis) and the amount of nandrolone decanoate (*x*-axis) were subjected to plot a calibration plot. Regression analysis of results was used to obtain a linear regression model. Figure 4 illustrates representative calibration plot *A* = 535.1 + 2155.5·*X* (*n* = 6).

A good linear relationship over the whole range of nandrolone decanoate amount confirms the linear regression data listed in Table 1. The correlation coefficient (r) of the fitted model is equal to 0.9998 and indicates a relatively strong relationship between variables. The significance level (*p*) of less than 0.0001 shows that there is statistically significant correlation between the peak area registered from the chromatogram and the amount of nandrolone decanoate at each level. Thus, the proposed method is in agreement with the requirements of the ICH guidelines important for the quantitative analysis of an active ingredient such as nandrolone decanoate in pharmaceuticals.

#### 2.2.3. Limits of Detection and Quantification

The limits of detection (LOD) and quantification (LOQ) of the proposed HPTLC-densitometric method were determined on the basis of the calibration plot prepared using the peak area measurements of examined nandrolone decanoate at the working range, i.e., 0.780–12.500 μg/spot. LOD and LOQ values were calculated according to Equations (1) and (2) by means of the standard deviation of the intercept and slope of the calibration curve.

This level of detection and quantification equal to 0.231 and 0.700 μg/spot, i.e., less than 1 μg/spot, confirms that the adopted HPTLC-densitometric method is sensitive enough for the determination of examined nandrolone decanoate in its pharmaceutical formulation. A lower level of detection and quantitation can be observed in case of the OPLC procedure recommended for the in-process purity control of nandrolone [16]. However, it should be noted that the proposed HPTLC-densitometric method has two advantages compared to the earlier reported methods: it is simple and cost effective. Single chromatographic development using the binary phase and quick densitometric detection instead of a visualizing reagent as reported in the earlier study was used at the method development [16].

#### 2.2.4. Accuracy

The accuracy of this method was determined based on the recovery study (%R) by the method of standard addition at three levels of concentration. Known amounts of pure standard of nandrolone decanoate corresponding to 50%, 100%, and 150% of content present in the tested drug sample were added. Each sample in the quantity of 5 μL was analyzed in triplicate under adopted chromatographic conditions. The peak area was measured at each concentration level. The results are demonstrated as recovery percentage of nandrolone decanoate in Table 2.

The recovery was found to be in the range of 98.2–100.3% for 50%, 100%, and 150% of pure standard of nandrolone decanoate added. The recovery results are within the acceptance range according to the ICH guidelines as well as the label claim of the studied pharmaceutical formulation.

The satisfactory average value of recovery of 99.0% confirms that there is no interference from any additives present in the examined drug formulation.

The high percentage of accuracy presented in this work is very similar to that obtained for RP-HPLC assay of nandrolone phenylpropionate in marketed formulations [6]. The satisfactory recovery result ensures that the proposed TLC-densitometric procedure is free from the interference of inactive ingredients in the studied injection formulation without the time-consuming sample cleaning or derivatization processes required in the case of more modern chromatographic techniques such as HPLC and GC-MS [3,4,5,6].

#### 2.2.5. Precision

The precision of the method was evaluated in terms of repeatability (intra-day precision) and also intermediate precision (inter-day precision) by analyzing samples of pharmaceutical formulation corresponding to 0.780, 1.560, and 3.120 μg/spot of nandrolone decanoate three times on the same day (intra-day precision) and by analyzing the same samples on three different days over a period of three weeks. Table 3 shows the results of precision studies which are expressed as the coefficient of variation of the measured peak area (%CV) of nandrolone decanoate in analyzed samples at three different concentrations. As shown in Table 3, the values of the coefficient of variation for the intra-day precision were 1.08, 1.68, and 1.17, respectively, and for the intermediated precision (inter-day precision), %CV values were 0.80, 0.97, and 1.18. The %CV levels of intra- and inter-day precision were less than 2% in all cases, which confirmed that there were no significant variations in the analysis of nandrolone decanoate at the applied concentrations. Therefore, the proposed method is considered to be precise.

Comparison of our results with those obtained by other authors by means of modern chromatographic techniques, i.e., HPLC and GC, indicates that the precision (coefficient of variation) of the RP-HPLC method to determine nandrolone phenylpropionate in different pharmaceutical formulations described in the literature is placed in a much narrower range: 0.218–0.875% for intra-day and 0.441–0.875% for inter-day precision [6]. However, worse results of precision (<15%) can be observed in the case of quantitation of nandrolone and its metabolites in biological samples by means of the gas chromatography-mass spectrometric method [4].

#### 2.2.6. Robustness

The robustness of the developed method was investigated by analyzing the effect of small, deliberate variations of chromatographic conditions on the peak area of the examined drug sample. Variable factors were volume of mobile phase (±5 mL), time of activation of chromatographic plates (±5 min), chamber saturation time (±5 min), development distance (±10 mm), and volume of sample spotting onto chromatographic plate (±1 μL). The robustness of the method was checked at the amount of 2.500 μg/spot in three replicates in each case. The results of %CV (coefficient of variation) of peak area and % assay of nandrolone decanoate obtained at deliberate changes of chromatographic analysis are summarized in Table 4. The data listed in Table 4 indicate no marked change in results. The overall %CV of peak area was found to be less than 2% for all parameters of robustness. Similarly, the results of % assay of nandrolone decanoate are satisfactory and varied from of 98.0 to 100.1. Both parameters of robustness indicate that the developed method is robust. It should be noted that the adopted HPTLC-densitometric procedure was to be unaffected by small changes of chromatographic conditions.

#### 2.2.7. Analysis of Pharmaceutical Formulation

The validated HPTLC-densitometric methodology combined with densitometry was applied for the quantitative determination of nandrolone decanoate in its commercially available solution for injection (50 mg/mL). Drug sample corresponding to 2.50 μg of nandrolone decanoate per spot was analyzed on a HPTLC plate using adopted mobile phase such as n-hexane-ethyl acetate in ratio 42.5:7.5 at the distance of 7.0 cm. The densitometric scanning was made at 245 nm. The average content of examined active constituent was determined based on measured peak areas. Table 5 shows the results of nandrolone decanoate assay in pharmaceutical formulation.

The analysis of the pharmaceutical formulation of nandrolone decanoate indicates drug content 50.5 mg/mL or 101.0% as compared to the label claim. It is in good agreement with the recommendation of the ICH guidelines, as well as the pharmacopoeial requirements in terms of the average content of nandrolone decanote as an active ingredient in the pharmaceutical formulation [18,19]. The low value of CV (<1%) confirms the suitability of the proposed HPTLC-densitometric method for routine laboratory quality control of nandrolone decanoate in marketed injection formulation.

## 3. Materials and Methods

### 3.1. Reagents and Apparatus

#### 3.1.1. Pure Standard

Nandrolone decanoate (17β-hydroxy-19-nor-4-androsten-3-one 17-decanoate) was pharmaceutical-grade substance and fulfills the requirements of the European Pharmacopoeia reference standard (Sigma-Aldrich, Steinheim, Germany).

#### 3.1.2. Sample

The pharmaceutical product in the form of solution for injection labelled to contain 50 mg of nandrolone decanoate in 1 mL was procured from a local pharmacy market.

#### 3.1.3. Solvents

All organic solvents used in this study were of analytical grade. n-Hexane and ethyl acetate as the mobile phase components were purchased from POCh (Gliwice, Poland). Ethanol 99.8% (for HPLC) used to prepare the standard and sample solutions was also purchased from POCh.

#### 3.1.4. Chromatographic Plates

The chromatographic analysis was carried out using glass HPTLC plates consisting of silica gel 60 CNF_254_ (size 10 cm × 10 cm) purchased from Merck (Darmstadt, Germany). The plates were activated for 20 min at temperature 110 °C before use.

#### 3.1.5. Apparatus

Densitometric scanning of obtained spots was performed using a CAMAG TLC Scanner 3 (Camag, Muttenz, Switzerland) operating in the absorbance mode and controlled by WinCATS software, version 1.4.1 (Camag, Muttenz, Switzerland). Deuterium and a tungsten lamp were used as a radiation source. Spectrum scan speed was kept at 100 nm/s. The chromatographic plates were scanned with the slit dimensions of 8.00 mm × 0.40 mm and scanning speed of 20 mm/s. Densitometric analysis of chromatograms was performed at 245 nm (±1 nm). Data resolution was 100 μm per step.

#### 3.1.6. Standard Solution

The standard stock solution of examined nandrolone decanoate (1 mg/mL) was prepared using ethyl alcohol as a solvent. Working solutions at different concentration were prepared by dilution by means of the same solvent. All solutions were spotted manually using micropipettes (5 μL, Camag, Muttenz, Switzerland) in quantity of 5 μL (10 mm from the lower edge and left edge of the plate). The distance between the spotted solutions was 15 mm in each case.

#### 3.1.7. Sample Preparation

The pharmaceutical sample at concentration of 0.50 mg/mL was prepared by dissolving the injection of nandrolone decanoate solution in ethyl alcohol. All solutions were stored at 2–8 °C until use. The solution was spotted manually using micropipettes (5 μL, Camag, Muttenz, Switzerland) on the HPTLC in quantity of 5 μL (10 mm from the lower edge and left edge of the plate) in each case. The distance between the spots was 15 mm. Development and quantification were performed under adopted chromatographic conditions.

### 3.2. Method Validation

The HPTLC-densitometric method was validated in accordance with the ICH guidelines in terms of specificity, linearity, limit of detection, limit of quantification, precision, accuracy, and robustness [18].

#### 3.2.1. Specificity

To confirm the specificity of the described HPTLC method, the R_F_ value of band and spectra of the standard solution of nandrolone decanoate and the sample (prepared from the examined pharmaceutical formulation) obtained under proposed chromatographic conditions were compared.

#### 3.2.2. Linearity

The linearity of this method was assessed by applying a standard solution of examined nandrolone decanoate at 6 concentration levels in the range from 0.156 μg/μL to 2.500 μg/μL (0.156, 0.312, 0.624, 1.250, 1.800, 2.500). A quantity of 5 μL of each working solution was spotted on HPTLC plates to achieve spots of 0.780, 1.560, 3.120, 6.250, 9.000, and 12.500 μg/spot. The plates were developed using adopted chromatographic conditions. The peak area of nandrolone decanoate at each concentration was recorded by means of a TLC densitometer at λ = 245 nm. To obtain the average standard calibration plot, each analysis was repeated 6 times. The linear regression model was used for peak area and the amount of nandrolone decanoate. Linearity was determined as the relationship of peak areas *A* [AU] to the amount of nandrolone decanoate *X* [μg/spot].

#### 3.2.3. Limit of Detection (LOD) and Limit of Quantification (LOQ)

Limits of detection and quantification were determined based on the standard deviation of intercept and the slope of the calibration plot (*A* = a*X* + b) obtained using peak area measurements of examined nandrolone decanoate at the working range. The following equations were applied in order to calculate the mean LOD and LOQ values, respectively:LOD = 3.3 × SD/*a*(1)
LOQ = 10 × SD/*a*(2) where SD is the standard deviation of intercept, and *a* represents the slope of the calibration plot.

#### 3.2.4. Precision

The precision of the newly developed method was tested by applying a 5 μL sample of nandrolone decanoate at 3 different concentration levels (0.156, 0.312, and 0.624 μg/μL) on a HPTLC plate in 3 replicates each to achieve the amounts of 0.780, 1.560, and 3.120 μg/spot. The precision was exhibited as %CV (coefficient of variation) of nandrolone decanoate peak area. The repeatability (intra-day precision) was checked by analyzing the samples 3 times during the same day. Intermediate precision (inter-day precision) was evaluated by analyzing the samples on 3 different days over a period of 3 weeks.

#### 3.2.5. Accuracy

The method accuracy was determined by recovery studies and expressed as percent recovered R (%) of nandrolone decanoate at each concentration level. This experiment was carried out by the standard addition method in order to check the recovery of nandrolone decanoate at different levels in the studied formulation, i.e.*,* injection solution. The appropriate amount of nandrolone decanoate standard powder corresponding to 50%, 100%, and 150% of the label claim had been added to the sample solution. The chromatograms were analyzed under optimized chromatographic conditions. Determination was performed in triplicate at each level.

#### 3.2.6. Robustness

The robustness of the proposed TLC-densitometric method was checked by evaluating the effect of small but deliberate changes of applied chromatographic conditions on the results, i.e., on the measured peak area of studied nandrolone decanoate. Small changes in the proposed HPTLC procedure, such as the saturation time for developing chamber (±5 min), the volume of mobile phase used (±5 mL), development distance (±10 mm), the temperature of activation of chromatographic plates (±10 °C), and the amount of spotted sample solution (±1 μL), were introduced. The robustness was determined by applying 5 μL of nandrolone decanoate at concentration 0.50 μg/μL on chromatographic plate to achieve the amount of 2.50 μg/spot. This experiment was conducted in 3 replicates for each examined chromatographic conditions change. Mean values of obtained robustness data were calculated.

### 3.3. Analysis of Pharmaceutical Formulation of Nandrolone Decanoate

The validated HPTLC method was successfully applied for quantitative determination of nandrolone decanoate in its commercially available solution for injection at the concentration of 50 mg/mL. A quantity of 5 μL of diluted sample solution at concentration of 0.50 μg/μL was spotted on the HPTLC plate for development using optimum mobile phase, i.e., n-hexane-ethyl acetate in ratio 42.5:7.5 at the distance of 7.0 cm. After development, the plate was dried at room temperature. Densitometric scanning was conducted at 245 nm. The analysis of the spiked sample was repeated 6 times. The amount of nandrolone decanoate in the tested injection formulation was determined on the basis of measured peak areas.

## 4. Conclusions

A new, simple, and cheap TLC-densitometric procedure with densitometric detection was developed for the determination of nandrolone decanoate in the pharmaceutical injection formulation. The method enables direct determination of nandrolone decanoate in UV light without the need of derivatization of the sample and the use of dying reagents like those in other reported methods. The proposed method was validated as per the ICH guidelines. Statistical data showed that the method is specific, precise, accurate, and robust. Satisfactory results of detection and quantification limits confirm its sensitivity. The content of nandrolone decanoate in the tested commercial injection formulation is in a good agreement with label claims, as well as the pharmacopoeial recommendations. This TLC-densitometric methodology may be successfully applied in routine laboratory quality control of nandrolone decanoate in the commercially available injection formulation.

## Figures and Tables

**Figure 1 molecules-24-00435-f001:**
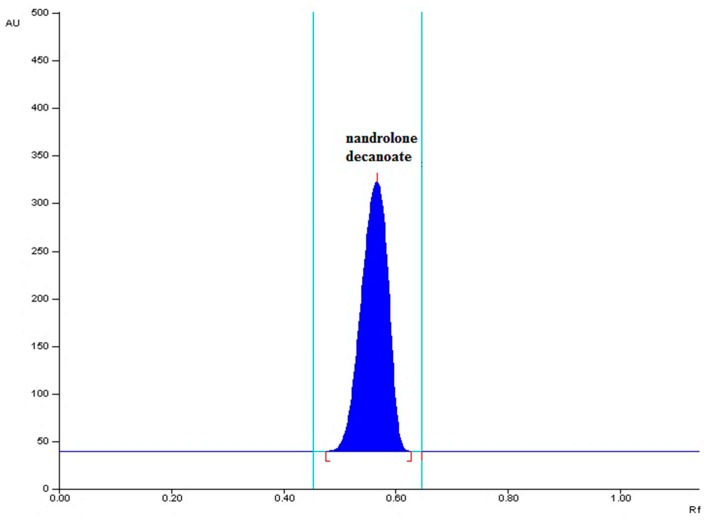
Representative densitogram of nandrolone decanoate spot of standard solution obtained using developed high-performance thin-layer chromatography (HPTLC) method at λ = 245 nm.

**Figure 2 molecules-24-00435-f002:**
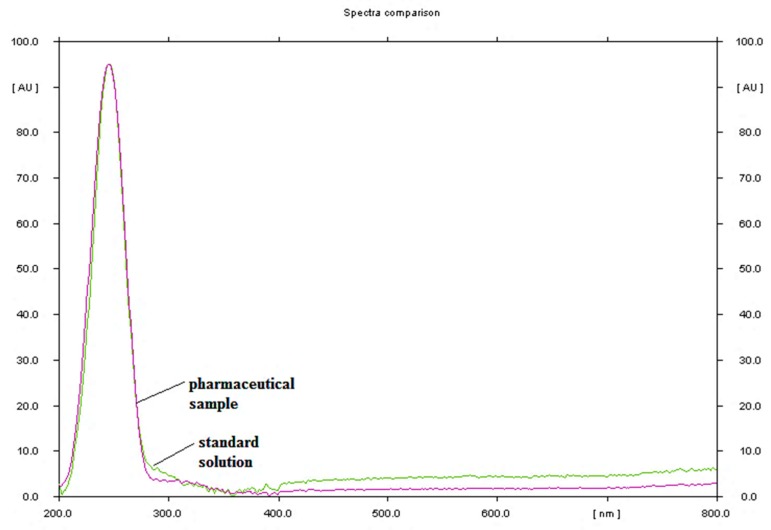
Absorption spectra of nandrolone decanoate standard solution and pharmaceutical sample obtained directly from HPTLC plates.

**Figure 3 molecules-24-00435-f003:**
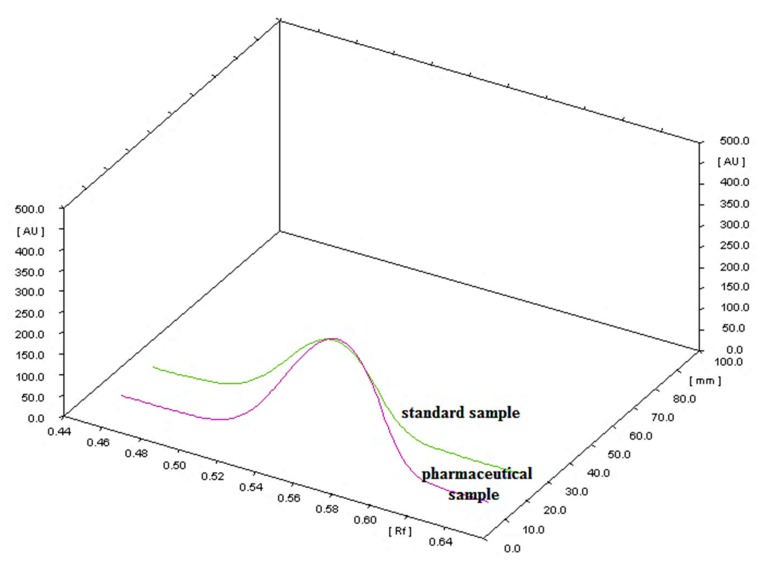
Three-dimensional HPTLC chromatogram showing nandrolone decanoate (standard) and pharmaceutical sample.

**Figure 4 molecules-24-00435-f004:**
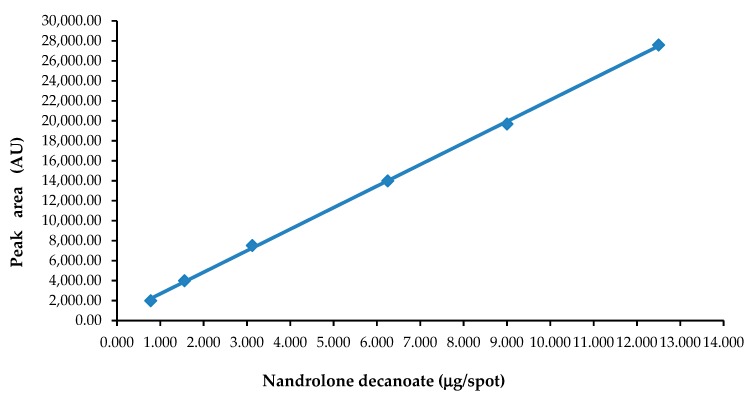
Calibration plot of nandrolone decanoate.

**Table 1 molecules-24-00435-t001:** Linear regression data of calibration plot.

Parameter	Results
Wavelength (nm)	245
Linearity range (μg/spot)	0.780–12.500
Slope	2155.5
Standard deviation of slope	21.8
Coefficient of variation (%CV) of slope	1.0
Intercept	535.1
Standard deviation of intercept	150.8
Correlation coefficient	0.9998
Standard deviation of residuals	222.9
F	9838.1
Significance level	<0.0001

**Table 2 molecules-24-00435-t002:** Accuracy data of proposed HPTLC-densitometric method for nandrolone decanoate pharmaceutical formulation. (SD = standard deviation; *n* = 3).

Initial Amount of Nandrolone Decanoate in Drug Sample (μg/μL)	% Amount Nandrolone Decanoate Standard Added	Theoretical Total Amount of Nandrolone Decanoate in Drug Sample (μg/spot)	Average Amount of Nandrolone Decanoate Recovered (μg/spot) ±SD	Recovery R (%)	Average Recovery R (%)
0.50	50	3.75	3.76 ± 0.21	100.3	99.0
0.50	100	5.00	4.93 ± 0.10	98.6	
0.50	150	6.25	6.14 ± 0.11	98.2	

**Table 3 molecules-24-00435-t003:** Precision of proposed HPTLC-densitometric method for nandrolone decanoate pharmaceutical formulation. (CV = coefficient of variance).

**Intra-Day Precision (Repeatability, *n* = 3)**		
**Amount of Nandrolone Decanoate (μg/spot)**	**Measured Peak Area ± SD**	**%CV of Peak Area**
0.780	2139 ± 23	1.08
1.560	3857 ± 65	1.68
3.120	7370 ± 86	1.17
**Inter-Day Precision (Intermediate Precision, *n* = 3)**		
**Amount of Nandrolone Decanoate (μg/spot)**	**Measured Peak Area ± SD**	**%CV of Peak Area**
0.780	2124 ± 17	0.80
1.560	3799 ± 38	0.97
3.120	7128 ± 84	1.18

**Table 4 molecules-24-00435-t004:** Robustness data of proposed HPTLC-densitometric method for nandrolone decanoate pharmaceutical formulation (*n* = 3). (CV = coefficient of variance; *n* = 3)

Variations of Chromatographic Conditions	%CV of Peak Area	% Assay of Nandrolone Decanoate
Volume of mobile phase (±5 mL)	1.07	98.8
Time of activation of chromatographic plates (±5 min)	0.76	98.0
Chamber saturation time (±5 min)	0.63	98.4
Development distance (±10 mm)	1.20	100.1
Volume of sample spotting onto chromatographic plate (±1 μL)	1.42	99.3

**Table 5 molecules-24-00435-t005:** Assay of nandrolone decanoate in pharmaceutical formulation.

Parameter	Data
Number of analyses	6
Label claim of nandrolone decanoate (mg/mL)	50.0
Average amount of nandrolone decanoate (mg/mL)	50.5
Minimum amount of nandrolone decanoate (mg/mL)	50.1
Maximum amount of nandrolone decanoate (mg/mL)	50.9
Standard deviation (SD)	0.4
Coefficient of variation (%CV)	0.8
Confidence interval of arithmetic mean with confidence level equal 95%	µ = 50.5 ± 0.4
Amount of nandrolone decanoate (%) in relation to label claim	101.0

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
