# Peer review of "Simple and Accurate HPTLC-Densitometric Method for Assay of Nandrolone Decanoate in Pharmaceutical Formulation"

_molecules, 2019, doi:10.3390/molecules24030435_

Round 1

Reviewer 1 Report

The authors describe the newly developed HPTLC method for determination of nandrolone decanoate in pharmaceutical formulations using typical normal-phase conditions on cyano- modified plates as a stationary phase. The method was validated to all major parameters according to the ICH guidelines i.e., selectivity, linearity, limits of detection and quantification, accuracy, precision (both repeatability and within laboratory reproducibility), and robustness. The article is nicely written, and the literature is adequately cited. There are only few papers describing the TLC methods for determination of nandrolone decanoate, therefore this work can be considered as a nice contribution to the scientific field of TLC of steroid compounds and is perfectly suited for the special issue (Steroids) of the Molecules journal.

Remarks:

Line 254: What was the distance from the plate edges (lower, left and right) as well as between the spots/lines of the standard and sample solutions applied to the plate. How was the application done – manually or using automated equipment (e.g. linomat)? Please describe in detail.  

Author Response

Response for Review 1

Remarks:

Line 254: What was the distance from the plate edges (lower, left and right) as well as between the spots/lines of the standard and sample solutions applied to the plate. How was the application done – manually or using automated equipment (e.g. linomat)? Please describe in detail.

Dear Reviewer,

We would like to thank you very much for preparing this review and all pertinent comments on our manuscript titled “Simple and Accurate HPTLC-densitometric Method for Assay of Nandrolone Decanoate in Pharmaceutical Formulation”. Based on your valuable comments concerning the lack of important datail about the sample and standard application on the applied chromatographic plates, we have revised our manuscript and introduced the needed data in Section 3.1.6 and Section 3.1.7 (page 8 and 9).

Below, we have presented the corrected sentence fragments (in yellow). We hope that it will meet with your requirements.

3.1.6. Standard solution

The standard stock solution of examined nandrolone decanoate (1 mg/mL) was prepared using ethyl alcohol as a solvent. Working solutions at different concentration were prepared by dilution by means of the same solvent. All solutions were spotted manually using micropipettes (5 mL, Camag, Muttenz, Switzerland) in quantity of 5 mL (10 mm from the lower edge and left edge of the plate). The distance between the spotted solutions was 15 mm in each case.

3.1.7. Sample preparation

The pharmaceutical sample at concentration of 0.50 mg/mL was prepared by dissolving the injection of nandrolone decanoate solution in ethyl alcohol. All solutions were stored at 2–8 °C until use. The solution was spotted manually using micropipettes (5 mL, Camag, Muttenz, Switzerland) in quantity of 5 mL (10 mm from the lower edge and left edge of the plate) in each case. The distance between the spots was 15 mm. Development and quantification were performed under adopted chromatographic conditions.

Sincerely yours,

The authors

Reviewer 2 Report

This is a well written manuscript with sound scientific data. While analytical standard methods for this compound already exist, the proposed method is very straight forward and seams to be easy to apply and is therefore of slight advantage over the existing methods.

However, the method is based on TLC and the authors don't mention the use of modern TLC instruments in regards to application and development. The reader gets the impression that the sample have been applied manually and it is very unlikely that the results will be reliable and reproducible if the samples are not applied with an automatic TLC sampler.

It is unclear if the method is able to distinguish between the analyte and metabolites of the analyte and if impurities can be detected.

Author Response

Response for Review 2

Comments and Suggestions for Authors

This is a well written manuscript with sound scientific data. While analytical standard methods for this compound already exist, the proposed method is very straight forward and seams to be easy to apply and is therefore of slight advantage over the existing methods. 

However, the method is based on TLC and the authors don't mention the use of modern TLC instruments in regards to application and development. The reader gets the impression that the sample have been applied manually and it is very unlikely that the results will be reliable and reproducible if the samples are not applied with an automatic TLC sampler. 

It is unclear if the method is able to distinguish between the analyte and metabolites of the analyte and if impurities can be detected. 

Dear Reviewer,

We would like to thank you very much for review and all pertinent comments on our manuscript. We fully agree with all the comments.

We agree with the comment that the automated TLC sampler is the needed tool for the application of sample solution on the chromatographic plates in current TLC analysis. What is also important, it allows to spot the sample solution in the wide range of volume and the achieved LOD values probably can be much lower.

Dear Reviewer in response to your valuable comments we would like to assure and explain that based on our experience/practice in the field of the TLC we tried to obtain all results with suitable precision and accuracy using the suitable for TLC analysis micropipettes (5 mL, Camag, Muttenz, Switzerland). It was characterized by validation parameters which are satisfactory enough for the purpose of pharmaceutical analysis of nandrolone decanoate in examined preparation. The applied volume of analyzed sample was constant (5 mL) in each case, thus enabled also to reduce the potential errors during performed analysis. As it was highlighted in this paper, all results are performed as the average values of three or six analyses, respectively. The proposed HPTLC method is dedicated for the assay of nandrolone decanoate in its simple pharmaceutical dosage form, namely injection solution, therefore in the presence of present additives e.g. impurities. According to literature review for the purpose of analysis the metabolites of studied steroid in biological samples the most recommended is HPLC as well as GC method.

The results of specificity study confirmed the purity of examined pharmaceutical preparation. There was no additional peaks on obtained densitogram.

Our work confirms that the proposed method can be applied for the quality control of simple pharmaceutical formulation of nandrolone decanoate. Till now there is a lack of serious TLC/HPTLC procedure for this purpose. The proposed procedure can be important especially in the case of the lack of HPLC or GC apparatus in the laboratory.

Dear Reviewer we hope that our response are in agreement with your requirements.

Sincerely yours,

The authors

Reviewer 3 Report

The authors present in their manuscript Simple and Accurate HPTLC-densitometric Method for Assay of Nandrolone Decanoate in Pharmaceutical Formulation.

Manuscript is well written but scientific novelty is very low. There is presented routine method for determination of one analyte in very simple matrix. In my opinion, the article should be submitted to more focused pharmaceutical journal. It can be published as validation protocol in Pharmacopoeia…..

Other reasons of my rejection:

Method of determination is too routine and too simple to be published.

There is no novelty from the analytical chemistry point of view. HPTLC is well established method.

There is no problem to be solved – one analyte, no separation, too simple matrix (injection solution).

The solution used for TLC are too toxic.

Organic solvents as acetone, n-hexane and ethyl acetate do not fulfill requirements for green chemistry approach in routine laboratory.

Scientific novelty of the whole manuscript is too low.

Can be the method automated for routine laboratories?

What is pros of HPTLC method in comparison with eg. HPLC?

In my opinion, manuscript should be submitted as application note to pharmaceutically focused journal.

Author Response

Response for Review 3

Dear Reviewer,

We would like to thank you very much for review and all pertinent comments on the manuscript titled: Simple and Accurate HPTLC-densitometric Method for Assay of Nandrolone Decanoate in Pharmaceutical Formulation.

Below, we tried to answer for all valuable comments, point by point of how each comment was addressed in the review.

We will hope that it will meet with your requirements.

Comments and Suggestions for Authors

The authors present in their manuscript Simple and Accurate HPTLC-densitometric Method for Assay of Nandrolone Decanoate in Pharmaceutical Formulation.

Comment 1

Manuscript is well written but scientific novelty is very low. There is presented routine method for determination of one analyte in very simple matrix. In my opinion, the article should be submitted to more focused pharmaceutical journal. It can be published as validation protocol in Pharmacopoeia…..

Authors response

Dear Reviewer thank you very much for this suggestion. In response to this comment we would like to explain that as it was highlighted in our paper till now there is no simple and cost-effective procedure like described HPTLC method for quality control of simple pharmaceutical formulation in form of solution for injection containing nandrolone decanoate. Simple form (simple matrix) is the main form which is used for medical use. This is the reason the developing by us the method of determination of nandrolone decanoate in simple matrix. For the purpose of analysis of this compound in biological samples (urine, blood) thus in complex matrix at lower level a proper HPLC and GC procedures are available accordance with cited Literature (Ref. 3-6).

Comment 2

Method of determination is too routine and too simple to be published.

Authors response

Dear Reviewer thank you very much for this comment. We agree with you that among various analytical techniques, TLC/HPTLC method is very well known and popular. In response to this suggestion we would like to explain that the current literature review indicates that despite of modern chromatographic techniques and new detection systems it is widely used in combination with densitometry for the purpose of pharmaceutical analysis of various active ingredients including the steroids in their simple as well as combined pharmaceutical formulations such as:

Ł. Komsta, M. Waksmundzka-Hajnos, J. Sherma. Thin Layer Chromatography in Drug Analysis. CRC Press , 2013.

J. Sherma. Review of HPTLC in Drug Analysis: 1999-2009. JAOAC Int. 93 (3), 2010.

O Sulaiman , R Hashim , M.N Mohamad Ibrahim.Thin-Layer Chromatographic Analysis of Steroids: A Review. Trop J Pharm Res, 9 (3), 301-313, 2010.

The main advantage of TLC/HPTLC technique with densitometry what is widely highlighted in the scientific papers is its simplicity, low cost in comparison with other new chromatographic techniques. What is also important the purity of sample can be not very high like in the case of other chromatographic methods. Because the amount of active ingredient (i.e. nandrolone) in pharmaceutical preparations is high, therefore the sensitivity of TLC/HPTLC in combination with densitometry is enough for quality control of them.

Comment 3

There is no novelty from the analytical chemistry point of view. HPTLC is well established method.

Authors response

Dear Reviewer thank you very much for this comment. In response to this suggestion we would like to explain that the aim of this work was to develop the simple, accurate and cost-effective procedure for the determination of nandrolone decanoate in commercially available preparation. It is simple pharmaceutical preparation in form of injection.

The lack of an official TLC/HPTLC with densitometry procedure for this purpose confirms the novelty of this work.

This procedure fulfills all ICH requirements. It can be used as for example an alternative method when there is a lack of more modern chromatographic apparatus.

Comment 4

There is no problem to be solved – one analyte, no separation, too simple matrix (injection solution).

Authors response

Dear Reviewer thank you very much for this comment. We agree with this suggestion. However in response to this comment we would like to explain that the aim of this work to find the chromatographic conditions (for HPTLC method) suitable for commercially available form of examined nandrolone decanoate, which is not yet available. The most widely used preparation is that consisted of one active compound, namely nandrolone decanoate. For this fact we have analyzed this simple matrix (injection solution).

Comments 5 and 6

The solution used for TLC are too toxic.

Organic solvents as acetone, n-hexane and ethyl acetate do not fulfill requirements for green chemistry approach in routine laboratory.

Authors response

Dear Reviewer thank you very much for this comment. We are very grateful for this very important comment and we agree with the comment that the applied solvents as mobile phase composition are not recommendable from an environmental perspective and human safety. In response to this comment we would like to justify the choice of these mobile phase components as the most popular for the purpose of steroids analysis in available literature. We decided to select this binary mobile phase because enabled to obtain satisfactory results of retardation factor value in the acceptable range for the purpose of HPTLC analysis using the applied CN plates, acceptable LOD, LOQ and other parameters.

Dear Reviewer based on your valuable comment we will be very careful in the choice of mobile phase components in our future studies.

Comment 7

Scientific novelty of the whole manuscript is too low.

Authors response

Dear Reviewer thank you very much for this comment. In response to this comment similarly like in above performed authors response we would like to indicate that this is the first developed HPTLC method dedicated directly to quality control of nandrolone decanoate in its simple commercially available preparation. Because it is simple and low cost procedure, therefore it can be used in each laboratory especially in that in which there is a lack of modern chromatographic apparatus (HPLC, GC).

Comment 8

Can be the method automated for routine laboratories?

Authors response

Dear Reviewer thank you very much for this comment. The proposed HPTLC-densitometric procedure can be more automated especially in the case of the application system, which probably can improve the sensitivity of the method needed for lower amount of examined steroid usually observed in biological samples.

Comment 9

What is pros of HPTLC method in comparison with eg. HPLC?

Authors response

Dear Reviewer thank you very much for this comment. In response to this comment we would like to explain that the HPLC method is characterized by better sensitivity than TLC/HPTLC. However in the case of nandrolone decanoate present in form of pharmaceutical formulation the level of detection and quantitation at micrograms achieved by proposed procedure is enough. The main advantage of described HPTLC procedure is simplicity and low cost in comparison with HPLC which usually requires the time consuming purification of sample before the suitable HPLC analysis.

Comment 10

In my opinion, manuscript should be submitted as application note to pharmaceutically focused journal.

Authors response

Dear Reviewer thank you very much for this comment. We hope that you find our response well. In response to this comment we would like to confirm that the goal of our paper is with the aim of the scope of the journal. The TLC/HPTLC technique is described many times in this journal (Molecules) in various aspects (isolation, quality, quantity analysis) of different synthetic as well as natural (from plant materials) active compounds in different matrix including pharmaceutical formulations. What is important for the purpose o pharmaceutical analysis.

Sincerely yours,

The authors

Round 2

Reviewer 3 Report

The scientific novelty of the whole work is too low.

no additional experiments were done...